# Evaluation of Student Satisfaction with Ubiquitous-Based Tests in Women’s Health Nursing Course

**DOI:** 10.3390/healthcare9121664

**Published:** 2021-11-30

**Authors:** Mi-Young An, Yun-Mi Kim

**Affiliations:** College of Nursing, Gachon University, Incheon 21936, Korea; clickamy@hanmail.net

**Keywords:** education, nursing student, tablet computer, online learning

## Abstract

Learning evaluation using ubiquitous-based tests may be essential during a public health crisis, such as the COVID-19 pandemic, during which theoretical classes and clinical practice are conducted online. However, students may not be as familiar with ubiquitous-based tests as they are with paper-based tests. This survey study aimed to evaluate students’ satisfaction with ubiquitous-based tests and compare the evaluation results of a paper-based test with that of a ubiquitous-based test in nursing education. For the midterm exam of the Women’s Health Nursing course, a paper-based test was conducted, while a ubiquitous-based test using a tablet computer was used for the final exam. The Ubiquitous-Based Test Usefulness and Satisfaction tool, which has a five-point Likert-type response scale, was employed to evaluate the post-test usefulness and satisfaction scores of the ubiquitous-based test. The mean score of the ubiquitous-based test usefulness was 4.01 ± 0.67. There was a significant difference in satisfaction levels between the ubiquitous-based and the paper-based test (t = −3.36, *p* = 0.001). Specifically, the evaluation scores were not affected by different evaluation methods. Study participants deemed the ubiquitous-based test highly useful and satisfactory, suggesting that such tests may be a future-oriented evaluation method, potentially replacing paper-based tests.

## 1. Introduction

Owing to the rapid expansion and popularity of mobile devices, the education sector is undergoing innovative changes to enable learning without limitations of time, place, and devices [1,2]. With technology allowing for the expansion of the use of various smart mobile devices (e.g., tablet computers, smartphones, etc.) for learning, the Ministry of Education of Korea has also been innovating the educational system by implementing smart education and conducting evaluations [3]. Particularly, the coronavirus disease 2019 (COVID-19) pandemic has transformed the educational environment, favoring online classes and exams [4].

Since 2011, the Ministry of Education of Korea has been promoting computer-based test (CBT) evaluations using smart technology [2]. The Korea Health Channel Licensing Examination Institute used ubiquitous-based tests (UBT)—in which testing, scoring, and grade management are performed using tablet computers—to conduct the Clinical Pathology Mock Tests [5] in 2011, and the Doctor National Examination Mock Tests [6] in 2016. It has also been using UBTs for the Emergency Medical Examination (level 1) National Exams since 2017 [7]. Additionally, starting with the written examination for the Doctor National Examination in 2022, the national examinations for doctors, such as dentists and doctors of oriental medicine, will change to CBT in 2023, in which questions are uploaded to individual computers with answers written directly on said computers [8].

In nursing education, clinical practical skill performance is essential [2]. Undoubtedly, in the National Nurse Examination, the practical performance evaluation is helpful in managing the quality of a nursing license [2]. In Korea, the number of people taking the National Nurse Examination has been increasing every year, rising to 21,511 in 2019 [9]. For comparison purposes, the National Examination for Doctors had 3318 applicants in 2019. It is difficult to evaluate applicants’ practical skills in the National Nurse Examination because student preparation time for the practical exam is more than six times that required for the national exam [5]. According to a study, the evaluation of practical skill performance requires the same number of evaluators as the number of applicants and that the involvement of evaluators in subjective evaluations can also lead to real-world difficulties in fairness based on factors such as location, budget, time, and manpower [10]. However, by using UBT, it is possible to assess applicants’ practical skills through simulation video evaluation, a move that enables the simultaneous assessment of a large number of candidates without the constraints of location and time allocation. There is no need for a test site and test proctor [11], and no test papers need to be printed. As such, testing using a tablet computer has financial benefits, such as reduction in printing and management costs, effective space utilization, and excellent mobility [9]. It also enhances security by preventing students from accessing educational materials or the Internet from their computers after the test begins [12]. If UBT, with these advantages, is applied, it would enable the construction of proficient health manpower in Korea. Another study demonstrated that evaluation using such smart technology allows for the use of questionnaires that depict scenarios in a form that comes closer to a clinical environment (e.g., through the use of multimedia platforms), which is difficult to implement in paper-based tests, and raises the level of evaluation, ultimately producing competent health care personnel [8]. In line with this opinion, one study remarked that the conversion of paper-based tests to UBTs is very important and could raise the quality of health and medical examinations in Korea to a new level [13]. Given that nursing education is an applied science in which the harmony of theory and practice is important [14], UBT may be an effective method to evaluate students’/applicants’ practical skills, since it allows for the realistic presentation of nursing problems (based on clinical cases) using multimedia.

Due to the recent COVID-19 pandemic, many universities and nursing colleges in the health care field are attempting to use paperless tests that can be conducted in a non-contact environment [12,15,16]. A lot of research is being carried out on this. In Korea, studies were conducted on tests using a tablet computer for medical [5,16,17,18] and emergency rescue students [19]. These studies showed high satisfaction among the examinees due to the convenience in the administration of tests using a tablet computer. Additionally, in a study [16] of medical students in Korea, the convenience of using a tablet-computer-based test showed positive results. However, it was found that students still preferred the traditional paper-based test (PBT). A study of nursing college students in Korea, Vietnam, and Mongolia [15] also showed that CBTs were cost-effective and convenient. Meanwhile, a study [20] of medical students in Germany showed that tablet-computer-based learning significantly improved students’ grades. In the case of the United States, a study of CBT was conducted targeting college [21], dental college [12], and pharmaceutical college students [22]. It was found that they felt comfortable with CBTs and preferred them over PBTs. Furthermore, in Iran, a study was conducted on the relationship between CBT and anxiety among nursing students [23]. The results showed that there was no difference in anxiety between the PBT and CBT.

As such, there are many positive research results in studies of tablet-computer-based tests and CBT, but these results also vary. In addition, compared with PBT, students may be unfamiliar with tests using a tablet computer. Therefore, research is needed to analyze nursing students’ satisfaction with UBTs and compare its evaluation results with those of PBT. This study aims to examine students’ satisfaction with UBTs and compare their evaluation results of these tests with those of PBT in nursing education. Using UBTs to evaluate students’/applicants’ learning outcomes will provide useful data that can open up new horizons in the field of nursing evaluation and improve the quality of nurses that are the closest to medical consumers [8].

## 2. Materials and Methods

### 2.1. Design

This study used a survey questionnaire to compare the satisfaction and evaluation results of UBT with those of PBT in nursing education.

### 2.2. Participants

The study participants included 66 second-year nursing students who were taking part in the Women’s Health Nursing course in a nursing department of Gachon University in Incheon, Korea. After explaining the study’s aims and methods, students who were willing to voluntarily participate had their data collected using a tablet computer.

The sample size was calculated using the G ∗ Power 3.1 software (Heinrich-Heine-Universität Düsseldorf, Düsseldorf, Germany). The commonly used effect size was Cohen’s medium effect size, which is appropriate with a power of 0.8 or more [14]. The appropriate sample size for the dependent *t*-test analysis with a medium effect of 0.5 (*d* = 0.5), a significance level of 0.05 (*p* = 0.05), and a power of 0.95 (1-β = 0.95) was determined to be 54 students. Therefore, considering the dropout rate, 66 students were selected as the final sample.

### 2.3. Definition of Terms

Computer-Based Testing (CBT): Computer-based testing (CBT) involves reading questions, responding to them on individual computers, and submitting them [17].Ubiquitous-Based Testing (UBT): In ubiquitous-based testing (UBT), testing, scoring, and grade management are performed using tablets and smartphones [17].

### 2.4. Research Tools

#### UBT Usefulness and Satisfaction

The Ubiquitous-Based Test Usefulness and Satisfaction tool used in the Graduate School of Medicine of Kyung Hee University 2012 was utilized in this study; it was modified and supplemented after receiving approval for use from the original author [17].

The original tool can be applied using a tablet computer and serves to compare students’ satisfaction and perceptions regarding the usefulness of the UBT with that of a paper-based test. It has 17 questions, which are classified into the following four subscales: proficiency in using smart devices, convenience of UBT, UBT usefulness as a medical education evaluation tool, and UBT impressions. The responses were scored on a 4-point scale (very much; somewhat; not so much; not at all), with higher scores suggesting greater satisfaction and more positive perceptions of usefulness. In the original study, Cronbach’s α of the tool was 0.87 in the pre-test and 0.93 in the post-test. The tools revised and developed in this study were subdivided into UBT satisfaction and paper-based test satisfaction. Thirteen questions were added to the 17 already present in the original tool, taking the total to 30 questions, which were categorized into the following subscales: UBT usefulness (2 questions), convenience of UBT in the problem-solving process (12 questions), UBT satisfaction (8 questions), and paper-based test satisfaction (8 questions). The responses were scored on a 5-point Likert-type scale (1 point, very dissatisfied; 5 points, strongly agree), with higher scores indicating greater satisfaction and more positive perceptions of usefulness regarding UBT.

To confirm the reliability and validity of the modified tool, an expert content-validity test and a pilot study were conducted. Three nursing professors and two UBT experts provided a content-validity index value for the modified tool, which was confirmed to be 0.93 (0.80–1.0). Additionally, all of the corrected item-total correlation values were higher than 0.5, indicating that the internal consistency of the questionnaire was quite good. Regarding validity, Kaiser-Meyer-Olkin (KMO) and Bartlett’s sphericity tests were applied. The KMO value was 0.835, indicating that Bartlett’s test of sphericity was very significant (*p* < 0.001). On 22 October 2018, the pilot study was conducted with 67 second-year nursing students during the midterm exam of the Women’s Health Nursing course, yielding a Cronbach’s α of 0.95; in the main study, the Cronbach’s α was 0.89.

### 2.5. Data Collection

Prior to commencing the research, approval was sought from the Gachon University Ethics Review Board. The questionnaire that was utilized contained items that comprised only data presentation, with questions containing videos deleted. The classes during the course and the examinations were conducted by the same professor for both the midterm and final exams. The final exam, which used a UBT, was administered to the same students that responded to the midterm exam. Data collection was conducted over two sessions: one during the midterm test and one during the final test. The final test was conducted using a tablet computer. The questionnaire was completed by the examinees on the tablet computer screen after they finished the test.

#### 2.5.1. Midterm Paper-Based Test

In order to compare the results of the paper-based test and the UBT results, the midterm scores that were calculated before the Gachon University Ethics Review Board approval were used retrospectively. This was approved by the second IRB. On 23 October 2018, the midterm exam for the Women’s Health Nursing course was conducted; the questions were provided using a paper-based questionnaire, and the answers were written on the optical mark read (OMR) cards. All 28 items were multiple-choice questions comprising only data presentation. The test was expected to be completed within a maximum of 50 min. The difficulty level of the items was 0.59, and the level of item discrimination was 0.33.

#### 2.5.2. Final UBT

On 18 December 2018, the final exam was conducted using a UBT through a tablet computer. All the students completed the test at the same place. A mobile relay server without a network was used, which increased the security of the evaluation. The relay server can be moved at any time. In addition, the tablet computer does not turn off, even in the event of a power outage. After the start of the test, the security and safety of the assessment was maximized by restricting students’ access to educational materials and the Internet on the tablet computer. In order to minimize selection bias in answers due to the time difference, the UBT was conducted immediately after the PBT in other subjects. Each student was provided with one tablet computer, and all students were informed of the method. The test began after the participants entered their certification number, student number, and name, and the administrator approved the test’s onset. The questions were checked and answered directly on the tablet computer. The test comprised 40 multiple-choice questions that contained only data presentation—no multimedia. The test was expected to be completed within a maximum of 50 min. When the test time was over, the test page closed automatically, and all the responses were spontaneously submitted. The difficulty level of the items was 0.62, and the level of item discrimination was 0.39.

After UBT administration, data were collected on students’ gender, age, preference for and grades in the Women’s Health Nursing course, UBT usefulness and satisfaction, and paper-based test satisfaction (Figure 1).

### 2.6. Data Analysis

The collected data were analyzed using the SPSS/WIN 25.0 (IBM Corp.: Armonk, NY, USA, 2017) Statistical Program. Means and standard deviations were calculated for participants’ general characteristics, satisfaction with the Women’s Health Nursing course, UBT usefulness and satisfaction, and paper-based test satisfaction. The difference between satisfaction scores for the paper-based test and the UBT was analyzed by paired *t*-tests. The comparison of the satisfaction score for the midterm paper-based test and the final UBT was analyzed by chi-square test. Differences in UBTs according to general characteristics were analyzed using the Mann-Whitney U test and one-way ANOVA, and post-test analysis was conducted using the Kruskal-Wallis test.

### 2.7. Ethical Considerations

The study was conducted only after obtaining approval (1044396-201810-HR-192-01, 23 November 2018) from the Gachon University Ethics Review Board in Incheon Metropolitan City.

After explaining the study’s aims and methods, the participants who were willing to voluntarily participate were included in the research. They were informed that they could withdraw their participation at any time without facing any penalty. To ensure anonymity, the collected data were subsequently coded and used solely for the purposes of this research. Students who had been taught by the research director were excluded from this research. Information regarding participation consent and the purpose of the study was explained to all participants. Written consent was obtained from all participants.

## 3. Results

### 3.1. Participants’ General Characteristics

The participants included 12 males (18.2%) and 54 females (81.8%). The average age was 21.02 ± 1.81 years. Their preference for the Women’s Health Nursing course was 3.89 ± 0.88, with 28.8% responding “strongly agree” (*n* = 19), 36.4% “agree” (*n* = 24), 30.3% “average” (*n* = 20), and 4.5% “disagree” (*n* = 3). Moreover, 98.5% of the participants (*n* = 65) used smart devices. Regarding the experience of using a smart device in tests, 33.3% (*n* = 22) responded that they had previous experience, and 66.7% (*n* = 44) responded that they had no such experience. Regarding experience of CBT, 77.3% (*n* = 51) responded that they had prior experience, and 22.7% (*n* = 15) said they had no such experience (Table 1).

### 3.2. UBT Usefulness

The mean score for UBT usefulness was 4.01 ± 0.67. The item, “The test proceeded well according to the precautions for test takers using a tablet computer,” showed the highest mean score, at 4.21 ± 0.71. The item, “I was satisfied with the use of the tablet computer that was provided as a test equipment,” showed the lowest mean score, at 3.82 ± 0.82.

The mean score for convenience of UBT in the problem-solving process was 4.22 ± 0.59. The item, “The function to check unanswered questions before submitting the answers was convenient,” showed the highest mean score, at 4.47 ± 0.61, while “Seeing one question on one screen was convenient,” showed the lowest mean score, at 3.97 ± 0.94 (Table 2).

### 3.3. Comparison of Sstisfaction Score for the Midterm Paper-Based Test and Final UBT

There was a significant difference between the satisfaction scores for the midterm paper-based test and for the final UBT (*p* = 0.001). Specifically, there were significant differences for the following items: convenience in the problem-solving process (*p* = 0.006), assistance in checking the amount of time available before test completion (*p* = 0.004), reduction in psychological burden (*p* = 0.042), reduction in eye fatigue (*p* = 0.001), items using materials (e.g., photos, sounds, and videos) with a sense of realism (*p* < 0.001), and achievement of learning goals (*p* = 0.003). However, there was no difference in convenience (*p* = 0.951) and test-taker ability evaluation (*p* = 0.794) between the paper-based test and the tablet computer test (Table 3).

### 3.4. Comparison of Sstisfaction Score for the Midterm Paper-Based Test and Final UBT by Participants’ General Characteristics

There were no significant differences in satisfaction scores for the midterm paper-based test by gender (Z = −1.31, *p* = 0.191), preference for the Women’s Health Nursing course (χ^2^ = 1.32, *p* = 0.516), and experience using a smart device in tests (Z = −1.33, *p* = 0.184).

For the final exam, the UBT satisfaction scores demonstrated no significant differences by gender (Z = −1.00, *p* = 0.313) or experience using a smart device in tests (Z = −0.07, *p* = 0.946). However, there was a significant difference in UBT satisfaction scores (χ^2^ = 9.22, *p* = 0.010) for preference for the Women’s Health Nursing course (Table 4).

## 4. Discussion

This study, in combination with existing literature, highlights the necessity of evaluating practical performance in nursing education and using smart technology to improve the quality of nursing in Korea. By providing data on students’ satisfaction with UBTs, the findings of the current study are expected to contribute to the efficiency of nursing education evaluation and learning support in Korea. Only a few studies have used UBT; therefore, CBT, which is similar to UBT, was used for comparison.

The results show that the mean score for UBT usefulness was 4.01 ± 0.67, with many of the items receiving positive scores, such as a score of 3.82 ± 0.82 for the question, “I was satisfied with the use of the tablet computer provided as a test equipment,” and a score of 4.21 ± 0.71 for “The test proceeded well according to the precautions for test takers using a tablet computer.” This result is consistent with prior research [17], which showed high scores to the question, “Are you good at various smart devices?” when evaluating a UBT. Furthermore, the current results can be deemed as similar to those in two prior studies, which described positive reactions from medical students to a UBT [14,15,16]. These reactions may be attributed to the generalized use of and familiarity with smart devices in Korean society.

The results show that respondents deemed the UBT highly convenient (4.22 ± 0.59), with the question “The function to check unanswered questions before submitting the answers was convenient” having the highest score, at 4.47 ± 0.61, and all questions related to the convenience of UBT having shown high scores. Through interviews, the respondents remarked that the use of UBT allowed for wrong answers to be erased by a simple touch, without needing erasers, and that revised responses could be written immediately after deleting the incorrect ones. These were potentially deemed convenient because they allowed the respondents to access usual functions found in smart devices while answering the UBT questions. These findings are in line with earlier research [17], which showed that 73.9% and 68.5% of students provided positive answers to the following questions, respectively: “Are smart devices convenient in this test?” and “Are you satisfied with the composition and completeness of the test?” Additionally, another study [5] showed that participants provided answers about the convenience of UBTs that comprised sentences like, “Because only one item is displayed on one screen, it is possible to make fewer mistakes.” Another study showed that students had a high satisfaction with the “convenience of correcting answers” in a UBT [24]. Therefore, the results of these studies, as well as the current findings, are considered similar. However, in the CBT, students believed that it was uncomfortable that they could not take notes on problems that required calculations [25]. This problem can be addressed by activating a tool, such as a calculator or a memo window, on the tablet computer during the test.

There was a significant difference regarding satisfaction with the paper-based test and that with the UBT (*p* = 0.001). Specifically, there were significant differences in the following topics: reduction in psychological burden (*p* = 0.042) and assistance in checking the amount of time available before test completion (*p* = 0.004). UBT can highlight unsolved problems and help students check the test time remaining. Additionally, when a student decides to answer a difficult test question later, the tablet used in UBT informs the student of the number of the unsolved question. This way of providing visual information is useful in reducing the psychological burden of exam completion time.

These results may have appeared because respondents were already familiar with various smart devices, so they did not feel hesitant about using the UBT. The findings of the current study are in line with a previous study [23] in which students remarked that the quality of the test on the tablet computer was better than that of the paper-based test and that evaluation using a tablet computer was more convenient than a paper-based test because there were no errors related to the use of the OMR card. Supporting this view, another study showed that students deemed [18] the ability to immediately check what question had yet to be solved good. A prior study [26] suggested that satisfaction is a positive factor for encouraging continuous learning. Therefore, it is assumed that the use of UBTs, which exhibited high satisfaction levels in the current study, will have a positive impact on nursing education, potentially more positive than that of paper-based tests. However, the current findings also differ from those in Kwon et al.’s study [17], which demonstrated that students preferred the existing paper-based test over the UBT; this difference may be attributed to the age differences between the samples of these two studies. Specifically, Kwon et al.’s study [17] was conducted on students at a graduate school of medicine with an average age of 29.2 ± 2.4 years; the current study involved students at a graduate school of nursing with an average age of 21.02 ± 1.81 years. Specifically, the participants in the current study were younger than the subjects in Kwon et al.’s study [12], denoting that they may have been more skilled in the use of smart devices.

Moreover, among the items concerning satisfaction, there was a significant difference in items using materials (e.g., photos, sounds, and videos) with a sense of realism (*p* < 0.001). Correspondingly, despite the deletion of items that contained videos from the final UBT (because they were not suitable) in the current study, students still showed high satisfaction with the items that contained multimedia, such as photos and sounds with a sense of realism. Furthermore, tablet computers usually have a screen that is quite large, allowing for users to freely adjust the font size or the position of the screen while completing a test. Since most participants had experiences using smart devices, they were likely to be somewhat proficient at watching videos on these devices. These results are consistent with those of previous studies [24,27], which showed that CBT is useful and has high user satisfaction, owing to comfortable image viewing.

This is supported by a study that found that knowledge that can be applied to various cases in the field of medical education should be acquired [17], as well as another study that demonstrated that subjects demonstrated more active learning during clinical practice after conducting a test using a tablet computer [5]. Learning with the help of images and videos shows higher learning results [23], and an online platform is the motivation for learning [27]. Accordingly, the current findings and prior literature denote that UBTs may be more appropriate for evaluation of practical nursing skills than paper-based tests. Although not statistically significant, UBT scored lower (3.27 ± 1.00) in terms of test-taker ability evaluation. It was unfamiliar because this was the first time evaluation was being conducted using smart devices, and its effect test scores may be one of the influencing factors [17].

Regarding UBT satisfaction according to participants’ general characteristics, there was a significant difference in students’ preference for the Women’s Health Nursing course. This denotes that participants who had a higher preference for the course showed lower negative reactions toward the new UBT, which may have led to a greater likelihood of increased satisfaction with the UBT. Accordingly, participants’ satisfaction with a given course may influence their satisfaction with the evaluation method for that course. However, further research is needed in the future to confirm these suppositions.

Studies of CBT have shown that gender, age, and previous experience with online courses or computer-based testing do not affect students’ perceptions of and comfort with CBT [19,25]. However, computer familiarity is a factor that may affect students’ performance in computer-based tests [18]. To prevent such problems, it is necessary to educate students on the test method and the use of smart devices prior to the test. Moreover, it is necessary to provide students an opportunity to practice before the test so that problems that may occur during the test can be anticipated and addressed. Nonetheless, for students who have become accustomed to online classes since the onset of COVID-19 pandemic, using a tablet computer has become routine [28,29]. The tablet computers used by UBT do not differ from tablet computers used by students for their studies; therefore, less than 10 min of education before the test may be sufficient.

During the CBT, students may experience technical errors in computer networks [30]. Additionally, cheating behaviors continue to increase in health care education, which emphasizes high ethical standards [30]. CBT testing can reduce the likelihood of cheating by randomizing questions and answers and disabling other programs during the test [12,16,30]. In the case of UBT using a tablet computer, since a separate operating server is used, the server can be moved at any time, and randomization of questions and answers can be implemented. In addition, the tablet computer does not turn off, even in the event of a power outage. In this study, we tried to minimize technical errors and strengthen security by using a portable relay server.

Although research on CBT is increasing, there are limitations in comparison and analysis because research on UBT is limited. According to the technology acceptance model, which is widely used for adaptive decision making on the Internet and in information technology fields, if perceived usefulness and perceived ease of use are high, the technology can be used easily [30,31,32,33]. Perceived usefulness refers to the belief that a new technology improves performance, while perceived ease of use refers to the degree to which a technology can be easily used. When an individual can accurately understand and solve how they accept a test using a tablet computer, it can be used as a complete medical education evaluation tool [16]. The results of this study show that the convenience and usefulness of using a tablet computer are high. When comparing satisfaction, UBT has higher satisfaction levels than PBT. These results suggest that the subjects’ perceived usefulness and perceived ease of use are high and that testing using a tablet computer can be easily accepted. Thus, since the participants displayed results that denote that they are ready to change from paper-based tests to UBTs, the latter may be a future-oriented evaluation method that can replace the former. An earlier study showed that UBTs can reduce the cost, shorten the time, and secure the reliability of the evaluation method [9,11]. Receiving feedback from the professor at the end of the test affects students’ learning ability and satisfaction [23,25]. If practical evaluations are performed using photos, sounds, and videos with a sense of realism, nursing performance can be evaluated with greater accuracy; this may enable the preparation of nurses equipped with greater skills and expertise. Additionally, replacing paper-based tests with UBTs may be useful in the context of global outbreaks of infectious diseases, such as severe acute respiratory syndrome (SARS) and COVID-19, which triggered transformations in the educational environment through the wide inclusion of online education and evaluation.

Some of the methodological limitations of this research are as follows. Although research is being actively conducted, there is a current gap in the literature concerning the studied topic. Since this study performed UBTs using tablet computers, the number of institutions and participants that were investigated was limited. In addition, one subject and one grade were evaluated. Therefore, these factors should be considered in future studies to improve the generalizability of the study. Additionally, since the survey was conducted immediately after UBT completion, there are limitations concerning generalization of the results. Students were asked to recall their satisfaction with the Women’s Health Nursing paper-based test conducted in the midterm after the final UBT exam. Therefore, the findings may be limited by a retrospective bias. In this study, UBT was conducted immediately after the paper-based test of other subjects in order to minimize any bias due to time difference.

Nevertheless, this study has great significance because it presents implications for future nursing education evaluation by attempting a nursing evaluation using a tablet computer. It also presents a reasonable method for the evaluation of nursing practice because this study demonstrates sufficient satisfaction with UBT. The findings of this study denote that future research is warranted to examine and develop new evaluation methods, as these can contribute to improvement in the quality of nursing in Korea by helping increase the efficiency of nursing education and learning support. In future research, it is necessary to actively utilize photos and videos in the development of test questions for patient care with various nursing needs in clinical settings. Development of the question bank platform will help generalize evaluations in school education using tablet computers. The universities that desire to use this service will be able to achieve improved convenience. In addition, we suggest a continuous large-scale iterative research project comprising more schools, students from different grades, and various subjects.

## 5. Conclusions

Nursing education, where the harmony of theory and practical skills is important, is evolving toward e-learning, mobile learning, ubiquitous learning, and smart learning, while the evaluation methods of national examinations in Korea is also evolving. Owing to global health crises triggered by infectious diseases such as SARS and COVID-19, switching to evaluations using UBTs may be necessary in order to provide students with contactless forms of learning, where online simulation and multimedia are implemented. Therefore, we suggest the need for more research to determine a new evaluation method suitable for the changing educational environment. Specifically, UBTs using video-based presentation of various nursing cases may be a future-oriented and contactless evaluation method applicable for the efficient acquisition of nursing knowledge.

## Figures and Tables

**Figure 1 healthcare-09-01664-f001:**
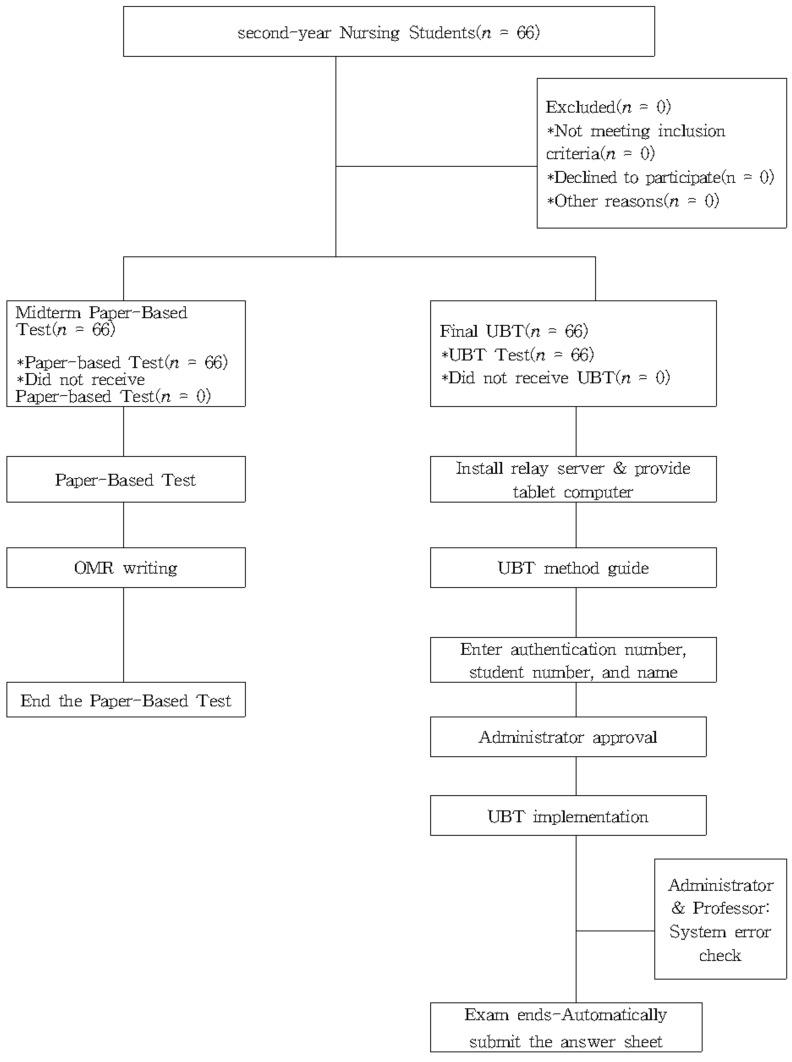
Flow diagram of the study. OMR = optical mark read; UBT = ubiquitous-based test.

**Table 1 healthcare-09-01664-t001:** General characteristics of nursing students (*n* = 66).

Variables	Categories	*n* (%)	M ± SD (Range)	Skewness	Kurtosis
Gender	Male	12 (18.2)			
	Female	54 (81.8)			
Age(yr)			21.02 ± 1.81 (19–29)	2.28	6.68
	≤0.22	56 (84.8)			
	≥0.23	10 (15.2)			
Preference with the Women’s Health Nursing course	Strongly Disagree		3.89 ± 0.88 (2–5)	−0.20	−0.89
	Disagree	3 (4.5)			
	Average	20 (30.3)			
	Agree	24 (36.4)			
	Strongly Agree	19 (28.8)			
Currently using a smart device (smartphones, tablet computers, etc.)	Yes	65 (98.5)			
	No	1 (1.5)			
Experience with testing using a smart devices (smart phones, tablet computers, etc.)	Yes	22 (33.3)			
	No	44 (66.7)			
Experience with testing using a computer (desktop)	Yes	51 (77.3)			
	No	15 (22.7)			

M = Mean; SD = Standard deviation.

**Table 2 healthcare-09-01664-t002:** Usefulness of UBT (*n* = 66).

Variables	Strongly Disagree n (%)	Disagree n (%)	Average n (%)	Agree n (%)	Strongly Agree n (%)	M ± SD
Usefulness of Smart Device						4.01 ± 0.67
I was satisfied with the use of the tablet computer that was provided as test equipment.		2(3.0)	23(34.8)	26(39.4)	15(22.7)	3.82 ± 0.82
The test proceeded well according to the precautions for test takers using a tablet computer.		1(1.5)	8(12.1)	33(50.0)	24(36.4)	4.21 ± 0.71
Convenience in the troubleshooting process for smart device.						4.22 ± 0.59
The reminder of the remaining time on the on-screen exam was more convenient than the examination time announcement broadcast.	1(1.5)	3(4.5)	6(9.1)	27(40.9)	29(43.9)	4.21 ± 0.90
The ability to select and correct answers was convenient.		2(3.0)	13(19.7)	30(45.5)	21(31.8)	4.06 ± 0.80
Seeing one question on one screen was convenient.		7(10.6)	9(13.6)	29(43.9)	21(31.8)	3.97 ± 0.94
The function to check unanswered questions before submitting the answers was convenient.			4(6.1)	27(40.9)	35(53.0)	4.47 ± 0.61
The previous problem, the next problem, and the way to see the whole problem was convenient.		2(3.0)	16(24.2)	20(30.3)	28(42.4)	4.12 ± 0.88
The ability to select and view test items, unexposed test items, checked items, and memo items was convenient.		1(1.5)	7(10.6)	28(42.4)	30(45.5)	4.32 ± 0.72
It was convenient to have a ‘check’ function to mark the problem to review later.		1(1.5)	9(13.6)	26(39.4)	30(45.5)	4.29 ± 0.76
The ‘wrong answer check’ function to check the wrong answer branch was convenient.		3(4.5)	11(16.7)	22(33.3)	30(45.5)	4.20 ± 0.88
The font size was appropriate.			5(7.6)	30(45.5)	31(47.0)	4.39 ± 0.63
The font was appropriate.		1(1.5)	5(7.6)	31(47.0)	29(43.9)	4.33 ± 0.68
It was convenient to view the data, such as enlarging/reducing the photo and watching the video again.		1(1.5)	13(19.7)	28(42.4)	24(36.4)	4.14 ± 0.78
The overall screen configuration was adequate.			9(13.6)	31(47.0)	26(39.4)	4.26 ± 0.68

M = mean; SD = standard deviation; UBT = ubiquitous-based test.

**Table 3 healthcare-09-01664-t003:** Comparison of satisfaction with midterm paper-based test and final UBT (*n* = 66).

Variables	Satisfaction of Paper-Based Test (OMR)	Satisfaction of UBT	t	*p*
	M ± SD	M ± SD		
Total Satisfaction	3.08 ± 0.66	3.60 ± 0.83	−3.36	0.001
Convenience in the problem-solving process	3.28 ± 1.01	3.88 ± 1.06	−2.84	0.006
Assistance in checking the amount of time available before test completion	3.06 ± 0.94	3.67 ± 1.08	−2.99	0.004
Reduction in psychological burden	2.86 ± 0.99	3.30 ± 1.20	−2.08	0.042
Reduction in eye fatigue	3.36 ± 1.01	3.98 ± 0.88	−3.63	0.001
Items using materials (e.g., photos, sounds, and videos) with a sense of realism	2.34 ± 0.83	3.76 ± 0.94	−9.52	<0.001
Convenience (paper-based Test (OMR) or tablet computer)	3.34 ± 1.17	3.36 ± 1.14	−0.62	0.951
Test-taker ability evaluation	3.31 ± 0.86	3.27 ± 1.00	0.26	0.794
Achievement of learning goals	3.06 ± 0.83	3.59 ± 0.97	−3.09	0.003

M = mean; OMR = optical mark read; SD = standard deviation; UBT = ubiquitous-based test.

**Table 4 healthcare-09-01664-t004:** Comparison of midterm exam paper-based test and final exam UBT according to general characteristics (*n* = 66).

Variables	Satisfaction of Paper-Based Test (OMR)			Satisfaction of UBT		
	M ± SD	Z or χ^2^	*p*	M ± SD	Z or χ^2^	*p*
		Gender				
Male (*n* = 12)	2.83 ± 0.48	−1.31	0.191	3.89 ± 0.88	−1.00	0.313
Female (*n* = 54)	3.13 ± 0.69			3.53 ± 0.81		
		Preference for the Women’s Health Nursing course				
Agree (*n* = 43)	3.02 ± 0.70	1.32	0.516	3.80 ± 0.86	9.22	0.010
Average (*n* = 20)	3.16 ± 0.59			3.31 ± 0.58		
Disagree (*n* = 3)	3.29 ± 0.75			2.62 ± 0.75		
		Experience using a smart device in tests				
Yes (*n* = 22)	3.26 ± 0.87	−1.33	0.184	3.66 ± 0.88	−0.07	0.946
No (*n* = 44)	2.99 ± 0.53			3.57 ± 0.81		

M = mean; OMR = optical mark read; SD = standard deviation; UBT = ubiquitous-based test.

## Data Availability

The data presented in this study are available on request from the authors.

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
