# Peer review of "Evaluation of Student Satisfaction with Ubiquitous-Based Tests in Women’s Health Nursing Course"

_healthcare, 2021, doi:10.3390/healthcare9121664_

Round 1
Reviewer 1 Report
This is a very interesting piece of research. It includes substantial literature review, justified methodology and draws interesting conclusions about the use of evaluation of Nursing Education in Higher Education institutes in Korea. The sample is large enough, the language is clear and it has appropriate referencing.

Author Response
[Thank you very much for your kind review.
We have modified the content as follows.]
- literature review
: It was written in Introduction(2-3page)
Due to the recent COVID-19 pandemic, many universities and nursing colleges in the health care field are attempting to use paperless tests that can be conducted in a non-contact environment [12, 15, 16]. A lot of research is being carried out on this. In Korea, studies were conducted on tests using a tablet computer for medical [5, 16, 18] and emergency rescue students [19]. These studies showed high satisfaction among the examinees due to the convenience in the administration of tests using a tablet computer. Additionally, in a study [20] of medical students in Korea, the convenience of using a tablet computer-based test showed positive results. However, it was found that students still preferred the traditional paper-based test(PBT). A study on nursing college students in Korea, Vietnam, and Mongolia [15] also showed that CBTs were cost-effective and convenient. Meanwhile, a study [21] on medical students in Germany showed that tablet computer-based learning significantly improved students’ grades. In the case of the United States, a study on CBTs was conducted targeting college [22], dental college [12], and pharmaceutical college students [23]. It was found that they felt comfortable with CBTs and preferred them over PBT. Also, in Iran, a study was conducted on the relationship between CBT and anxiety among nursing students [24]. The results showed that there was no difference in anxiety between the PBT and CBT. As such, there are many positive research results in studies of tablet computer-based tests and CBTs, but these results also vary.
- limitations(13-14page)
- Some of the methodological limitations of this research are as follows. Although research is being actively conducted, there is a current gap in the literature about the studied topic. Since this study performed UBTs using tablet computers, the number of institutions and participants that were investigated were limited. In addition, one subject and one grade were evaluated. Therefore, these factors should be considered in future studies to improve the generalizability of the study. Additionally, since the survey was conducted immediately after UBT completion, there are limitations in generalizing the results. Students were asked to recall their satisfaction with the Women's Health Nursing paper-based test conducted in the midterm after the final UBT exam. Therefore, the findings may be limited by a retrospective bias. In this study, UBT was conducted immediately after the paper-based test of other subjects, to minimize any bias due to time difference.
- Nevertheless, this study has great significance because it presents implications for future nursing education evaluation by attempting a nursing evaluation using a tablet computer. It is also significant as a reasonable evaluation method for evaluating nursing practice because this study demonstrated sufficient satisfaction with UBT. The findings of this study denote that future research is warranted to examine and develop new evaluation methods, as these can contribute to improving the quality of nursing in Korea by helping increase the efficiency of nursing education and learning support. In future research, it is necessary to actively utilize photos and videos in developing test questions for patient care with various nursing needs in clinical settings. Developing the question bank platform will help generalize evaluations in school education with using tablet computers. The universities that desire to use this service will be able to achieve the convenience.In addition, we suggest a continuous large-scale iterative research comprising of more schools, students from different grades, and various subjects.
Reviewer 2 Report
This paper is very interesting but is crucial that The authors describe the negative effe te of this IMO del ok the comunication and emotional and sensorial in particular. Can The authors describe the role of this in the comunication.
Author Response
[Reviewer 2]
This paper is very interesting but is crucial that The authors describe the negative effe te of this IMO del ok the comunication and emotional and sensorial in particular. Can The authors describe the role of this in the comunication.
[Thank you very much for your kind review.
We have modified the content as follows.]
<Revise>
- Introduction.(3page)
Meanwhile, a study [21] on medical students in Germany showed that tablet computer-based learning significantly improved students’ grades. In the case of the United States, a study on CBTs was conducted targeting college [22], dental college [12], and pharmaceutical college students [23]. It was found that they felt comfortable with CBTs and preferred them over PBT. Also, in Iran, a study was conducted on the relationship between CBT and anxiety among nursing students [24]. The results showed that there was no difference in anxiety between the PBT and CBT.
- Discussion (13page)
An earlier study showed that UBTs can reduce the cost, shorten the time, and secure the reliability of the evaluation method [9, 11]. Receiving feedback from the professor after the end of the test affects students' learning ability and satisfaction [26, 27]. If practical evaluations are performed using photos, sounds, and videos with a sense of realism, nursing performance can be evaluated with greater accuracy; this may enable the preparation of nurses equipped with higher skills and expertise. Additionally, replacing paper-based tests with UBTs may be useful in the context of global outbreaks of infectious diseases, such as severe acute respiratory syndrome (SARS) and COVID-19, that triggered transformations in the educational environment through the wide inclusion of online education and evaluation.
Reviewer 3 Report
I think that does not provide new evidence and isn´t interesting.
Specific comments:
- Writing
The writing, structure and organization of the manuscript is in accordance with the guidelines.
- Title
The title reflects the content and problem studied. Maybe, The title should contain the kind of study
- Abstract
The abstract reflects the manuscript. The methods is not clear
The Abstract should be improve.
- Key Words
The keywords are representative of the subject studied and exposed.
- Background
The background reflects the state of the art in relation to the study. The objective of the study is mentioned, as well as the justification for the choice and importance of studying this theme.
- Methods
There isn´t detailed description of the research methods used.
The design isn´t explained. Comparative isn´t a kind of research
They have described the setting and locations: They should describe more clear relevant dates and including periods of recruitment.
Eligibility criteria, sources and methods of selection of participants are not stated.
the explanation of statistical methods should be improved!! normality is calculated? that justifies the use of non-parametric measures?
Authors don´t describe any efforts to address potential sources of bias
- Findings
The results shown are concrete and detailed, explaining how to obtain this information and what scientific evidence it has.
The number of persons in each phase of the study is reported.
- Discussion
The key results of the discussion are concrete. In addition, it includes the main strengths and weaknesses in relation to other studies, discussing important differences in the results.
Limitations should be improved
- Conclusion
It´s clear and concise. the conclusions are in line with the objective.
- References
The references used are correct, the vast majority dating back less than ten years.
Author Response
[Thank you very much for your kind review.
We have modified the content as follows.]
<Revise>
- Abstract (1page)
This survey study aimed to evaluate students’ satisfaction with ubiquitous-based tests, and compare the evaluation results of a paper-based test with that of a ubiquitous-based test, in nursing education. In the midterm exam of the Women's Health Nursing course, a paper-based test was conducted, while a ubiquitous-based test, using a tablet computer, was used in the final exam.
- Methods (3, 6page)
- Design
This study used a survey questionnaire to compare the satisfaction and evaluation results of UBT with PBT in nursing education.
- Data Analysis
Differences in UBTs according to general characteristics were analyzed using the Mann-Whitney U test and one-way ANOVA, and post-test analysis was conducted using the Kruskal-Wallis test.
- Discussion(13-14page)
- Some of the methodological limitations of this research are as follows. Although research is being actively conducted, there is a current gap in the literature about the studied topic. Since this study performed UBTs using tablet computers, the number of institutions and participants that were investigated were limited. In addition, one subject and one grade were evaluated. Therefore, these factors should be considered in future studies to improve the generalizability of the study. Additionally, since the survey was conducted immediately after UBT completion, there are limitations in generalizing the results. Students were asked to recall their satisfaction with the Women's Health Nursing paper-based test conducted in the midterm after the final UBT exam. Therefore, the findings may be limited by a retrospective bias. In this study, UBT was conducted immediately after the paper-based test of other subjects, to minimize any bias due to time difference.
- Nevertheless, this study has great significance because it presents implications for future nursing education evaluation by attempting a nursing evaluation using a tablet computer. It is also significant as a reasonable evaluation method for evaluating nursing practice because this study demonstrated sufficient satisfaction with UBT. The findings of this study denote that future research is warranted to examine and develop new evaluation methods, as these can contribute to improving the quality of nursing in Korea by helping increase the efficiency of nursing education and learning support. In future research, it is necessary to actively utilize photos and videos in developing test questions for patient care with various nursing needs in clinical settings. Developing the question bank platform will help generalize evaluations in school education with using tablet computers. The universities that desire to use this service will be able to achieve the convenience.In addition, we suggest a continuous large-scale iterative research comprising of more schools, students from different grades, and various subjects.
Reviewer 4 Report
Being published in an international magazine, I would like you to refer to whether the model you are examining has been implemented in other countries. The calculation of the necessary sample on the total number of students of the institution may be consistent, but I believe that a greater number of centers is necessary to establish the conclusions that are expressed later. It is necessary to mention the ethics committee code and date of approval. The informed consent of the students, and justify why students are selected from 2 and not from other courses. I see a need for a more in-depth analysis of the psychometric properties of the tests, by researchers and their use. In the analysis of the sample, possible selection biases must be indicated. The results and statistical analysis, it is scarce if the researchers want to give greater scientific rigor to the work, they must do a statistical analysis in greater depth and rigor, the results seem preliminary and omitting some important elements. The discussion must follow the order of the results and have current citations that provide the most recent knowledge in favor or against its results, it is clear after reading in depth that this is not fulfilled. Check the bibliography citations older than 5 years are outdated, they should redo it entirely with a more up-to-date search.Author Response
[Thank you very much for your kind review.
We have modified the content as follows.]
<Revise>
- Introduction (2page)
Due to the recent COVID-19 pandemic, many universities and nursing colleges in the health care field are attempting to use paperless tests that can be conducted in a non-contact environment [12, 15, 16]. A lot of research is being carried out on this. In Korea, studies were conducted on tests using a tablet computer for medical [5, 16, 18] and emergency rescue students [19]. These studies showed high satisfaction among the examinees due to the convenience in the administration of tests using a tablet computer. Additionally, in a study [20] of medical students in Korea, the convenience of using a tablet computer-based test showed positive results. However, it was found that students still preferred the traditional paper-based test(PBT). A study on nursing college students in Korea, Vietnam, and Mongolia [15] also showed that CBTs were cost-effective and convenient. Meanwhile, a study [21] on medical students in Germany showed that tablet computer-based learning significantly improved students’ grades. In the case of the United States, a study on CBTs was conducted targeting college [22], dental college [12], and pharmaceutical college students [23]. It was found that they felt comfortable with CBTs and preferred them over PBT. Also, in Iran, a study was conducted on the relationship between CBT and anxiety among nursing students [24]. The results showed that there was no difference in anxiety between the PBT and CBT.
- Ethical Considerations (7page)
The study was conducted only after obtaining approval (1044396-201810-HR-192-01, 2018.11. 23) from the Gachon University Ethics Review Board in Incheon Metropolitan City.
Information regarding consent to participate and the purpose of the study was explained to all participants. Written consent was obtained from all participants.
- Research Tools(4page)
To confirm the reliability and validity of the modified tool, an expert content validity test and a pilot study were conducted. Three nursing professors and two UBT experts provided a content validity index value for the modified tool, which was confirmed to be 0.93 (0.80-1.0). Also, all of the corrected item-total correlation values were higher than .5, indicating that the internal consistency of the questionnaire was quite good. Regarding validity, Kaiser-Meyer-Olkin (KMO) and Bartlett's sphericity test were applied. The KMO value was .835, indicating that Bartlett's test of sphericity was very significant (p<.001). On October 22, 2018, the pilot study was conducted with 67 second-year nursing students during the midterm exam in the Women’s Health Nursing course, yielding a Cronbach’s α of 0.95; in the main study, the Cronbach’s α was 0.89.
- Data Collection (5page)
: After the start of the test, the security and safety of the assessment was maximized by restricting students’ access to educational materials and the Internet on the tablet computer. In order to minimize selection bias in answers due to the time difference, the UBT was conducted immediately after the PBT in other subjects.
- Discussion(13-14page)
- Some of the methodological limitations of this research are as follows. Although research is being actively conducted, there is a current gap in the literature about the studied topic. Since this study performed UBTs using tablet computers, the number of institutions and participants that were investigated were limited. In addition, one subject and one grade were evaluated. Therefore, these factors should be considered in future studies to improve the generalizability of the study. Additionally, since the survey was conducted immediately after UBT completion, there are limitations in generalizing the results. Students were asked to recall their satisfaction with the Women's Health Nursing paper-based test conducted in the midterm after the final UBT exam. Therefore, the findings may be limited by a retrospective bias. In this study, UBT was conducted immediately after the paper-based test of other subjects, to minimize any bias due to time difference.
- Nevertheless, this study has great significance because it presents implications for future nursing education evaluation by attempting a nursing evaluation using a tablet computer. It is also significant as a reasonable evaluation method for evaluating nursing practice because this study demonstrated sufficient satisfaction with UBT. The findings of this study denote that future research is warranted to examine and develop new evaluation methods, as these can contribute to improving the quality of nursing in Korea by helping increase the efficiency of nursing education and learning support. In future research, it is necessary to actively utilize photos and videos in developing test questions for patient care with various nursing needs in clinical settings. Developing the question bank platform will help generalize evaluations in school education with using tablet computers. The universities that desire to use this service will be able to achieve the convenience.In addition, we suggest a continuous large-scale iterative research comprising of more schools, students from different grades, and various subjects.
- References (14page)
: References have been corrected.
Reviewer 5 Report
Learning evaluation using computer-based tests has been seen as essential during a public health crisis, such as the COVID-19 pandemic, as theoretical classes and clinical practice have been conducted online.
As student are familiar with mobile devices, ubiquitous-based tests can be supposed to be more accessible (and user friendly) than computer-based tests, even if students are less familiar with this. However, students may not be as familiar with ubiquitous-based learning as they are with paper-based tests or even for computer-based tests for some of them.
This paper has evaluated students’ satisfaction with ubiquitous-based tests, after comparing the evaluation results of paper-based test versus ubiquitous-based test, in a nursing education context.
As a result, study participants felt that ubiquitous testing was very useful and satisfying, and that such testing can be a future-oriented assessment method, potentially replacing paper-based testing.
Needed corrections in text:
In the Introduction, "1.1Purpose" is not necessary as title and number here but the text after yes.
The same for 2.1 and 2.2, that can be removed, furthermore 2.2 is used twice. Numbering can be applied for the following.
Author Response
[Thank you very much for your kind review.
We have modified the content as follows.]
<Revise>
- Purpose has been modified. (3page)
Therefore, research is needed to analyze nursing students’ satisfaction with UBTs and compare its evaluation results with paper-based tests. This study aims to examine students’ satisfaction with UBTs and compare their evaluation results of these tests with those of PBT in nursing education. Using UBTs to evaluate students’/applicants’ learning outcomes will provide useful data that can open up new horizons in the field of nursing evaluation and improve the quality of nurses that are the closest to medical consumers [8].
- The number has been corrected. (3-4page)
: 2. Materials and Methods
2.1. Design
2.2. Participants
2.3. Definition of terms
2.4. Research Tools
2.4.1. UBT Usefulness and Satisfaction
Reviewer 6 Report
Dear Authors,
Research is extremely relevant for the technological advancement of teaching and learning in the field of nursing; however, some adjustments are suggested for the approval and publication of the manuscript.
I suggest including in the introduction, the state of the art in the use of Ubiquitous tests in other countries around the world. Highlight the financial impact and gains for the learning of students and educational institutions in the field of nursing.
Include signature of the Informed Consent Form of students who agreed to participate in the research.
The discussion could cover studies in different cultures on the researched topic, in addition, it would be pertinent to associate the characterization of the participants with the acceptance of technology, using, for example, the assessment of acceptance through models such as the technology acceptance model. (TAM) is an information systems theory that models how users accept and use technology. Present study limitations, contributions and innovation to the field of nursing education, thinking that nursing, like other courses in the health area, does not do without face-to-face teaching and contact with patients, as well as socialization with other professionals of the nursing team - which is the position of the authors of the manuscript in relation to the importance of face-to-face teaching and where the ubiquitous tests are inserted in this context. Include perspectives for future studies in the research area.
Observe 75% of references must be published in the last 5 years.

Author Response
[Thank you very much for your kind review.
We have modified the content as follows.]
<Revise>
- Introduction (2page)
- There is no need for a test site and test proctor [11], and no test papers need to be printed. As such, testing using a tablet computer has financial benefits such as reduction of printing and management costs, effective space utilization, and excellent mobility [9]. It also enhances security by preventing students from accessing educational materials or the Internet from their computers after the test begins [12]. If UBT with these advantages is applied, it would enable the construction of proficient health manpower in Korea.
- Due to the recent COVID-19 pandemic, many universities and nursing colleges in the health care field are attempting to use paperless tests that can be conducted in a non-contact environment [12, 15, 16]. A lot of research is being carried out on this. In Korea, studies were conducted on tests using a tablet computer for medical [5, 16, 18] and emergency rescue students [19]. These studies showed high satisfaction among the examinees due to the convenience in the administration of tests using a tablet computer. Additionally, in a study [20] of medical students in Korea, the convenience of using a tablet computer-based test showed positive results. However, it was found that students still preferred the traditional paper-based test(PBT). A study on nursing college students in Korea, Vietnam, and Mongolia [15] also showed that CBTs were cost-effective and convenient. Meanwhile, a study [21] on medical students in Germany showed that tablet computer-based learning significantly improved students’ grades. In the case of the United States, a study on CBTs was conducted targeting college [22], dental college [12], and pharmaceutical college students [23]. It was found that they felt comfortable with CBTs and preferred them over PBT. Also, in Iran, a study was conducted on the relationship between CBT and anxiety among nursing students [24]. The results showed that there was no difference in anxiety between the PBT and CBT.
- Ethical Considerations (7page)
Information regarding consent to participate and the purpose of the study was explained to all participants. Written consent was obtained from all participants.
- Discussion (13-14page)
- Although research on CBT is increasing, there are limitations in comparison and analysis because research on UBT is limited. According to the Technology Acceptance Model, which is widely used for adaptive decision-making on the Internet and in information technology fields, if the perceived usefulness and perceived ease of use are high, the technology can be used easily [32, 33, 34, 35]. Perceived usefulness refers to the belief that a new technology improves performance, while perceived ease of use refers to the degree to which a technology can be easily used. When an individual can accurately understand and solve how they accept a test using a tablet computer, it can be used as a complete medical education evaluation tool [20]. The results of this study showed that the convenience and usefulness of using the tablet computer were high. When comparing satisfaction, UBT had higher satisfaction levels than PBT. These results suggest that the subjects’ perceived usefulness and perceived ease of use are high and that testing using a tablet computer can be easily accepted.
- Some of the methodological limitations of this research are as follows. Although research is being actively conducted, there is a current gap in the literature about the studied topic. Since this study performed UBTs using tablet computers, the number of institutions and participants that were investigated were limited. In addition, one subject and one grade were evaluated. Therefore, these factors should be considered in future studies to improve the generalizability of the study. Additionally, since the survey was conducted immediately after UBT completion, there are limitations in generalizing the results. Students were asked to recall their satisfaction with the Women's Health Nursing paper-based test conducted in the midterm after the final UBT exam. Therefore, the findings may be limited by a retrospective bias. In this study, UBT was conducted immediately after the paper-based test of other subjects, to minimize any bias due to time difference.
- Nevertheless, this study has great significance because it presents implications for future nursing education evaluation by attempting a nursing evaluation using a tablet computer. It is also significant as a reasonable evaluation method for evaluating nursing practice because this study demonstrated sufficient satisfaction with UBT. The findings of this study denote that future research is warranted to examine and develop new evaluation methods, as these can contribute to improving the quality of nursing in Korea by helping increase the efficiency of nursing education and learning support. In future research, it is necessary to actively utilize photos and videos in developing test questions for patient care with various nursing needs in clinical settings. Developing the question bank platform will help generalize evaluations in school education with using tablet computers. The universities that desire to use this service will be able to achieve the convenience.In addition, we suggest a continuous large-scale iterative research comprising of more schools, students from different grades, and various subjects.
- References (14page)
: References have been corrected.
Round 2
Reviewer 3 Report
The manuscript has been improved and is now methodologically better.
Author Response
Thank you so much for your review.
Reviewer 4 Report
The changes have been made correctly, even so the quality and size of the sample seem insufficient to me.
Author Response
Thank you for your review.